# Scoring-Aggregating-Planning: Learning task-agnostic priors from interactions and sparse rewards for zero-shot generalization

## Abstract

Humans can learn task-agnostic priors from interactive experience and utilize the priors for novel tasks without any finetuning. In this paper, we propose Scoring-Aggregating-Planning (SAP), a framework that can learn task-agnostic semantics and dynamics priors from arbitrary quality interactions with sparse reward and then plan on unseen tasks in zero-shot condition. The framework finds a neural score function for local regional state and action pairs that can be aggregated to approximate the quality of a full trajectory; moreover, a dynamics model that is learned with self-supervision can be incorporated for planning. Many previous works that leverage interactive data for policy learning either need massive on-policy environmental interactions or assume access to expert data while we can achieve the similar goal with pure off-policy imperfect data. Instantiating our framework results in a generalizable policy to unseen tasks. Experiments demonstrate that the proposed method can outperform baseline methods on a wide range of applications including gridworld, robotics tasks and video games. [1]

## 1 Introduction

While deep Reinforcement Learning (RL) methods have shown impressive performance on video games (Mnih et al., 2015) and robotics tasks (Schulman et al., 2015; Lillicrap et al., 2015), they solve each problem *tabula rasa*. Hence, it will be hard for them to generalize to new tasks without re-training even due to small changes. However, humans can quickly adapt their skills to a new task that requires similar priors *e.g.* physics, semantics, affordances to past experience. The priors can be learned from a spectrum of examples ranging from perfect demonstrative ones that accomplish certain tasks to aimless exploration.

A parameterized intelligent agent "Mario" which learns to move to the right in the upper level in Figure 1 would fail to transfer the priors from to the lower level in Figure 1 and further play the game in the new level because change of configurations and background, *e.g.* different shapes of ladder, new fence. When an inexperienced human player is controlling the Mario to move it to the right in the upper level, it might take many trials for him/her to realize the falling to a pit and approaching the "koopa"(turtle) from the left are harmful while standing on the top of the "koopa"(turtle) is not. However, once learned, s/he can infer similar mechanisms in the lower level in Figure 1 without additional trials because human have a variety of priors including the concept of object, similarity, semantics, affordance, etc (Gibson, 2014; Dubey et al., 2018). In this paper, we intend to teach machine agents to realize and utilize useful priors to generalize to new tasks without finetuning.

Toward addressing the generalization problem with learned priors, we follow the intuition that: (1) each trajectory in a video game or a robotics task is composed of state-action pairs with object interactions (2) terminal rewards can be approximated by the aggregation of the scores for each state-action pairs. With those intuitions in mind, we summarize our proposed method in broad strokes. Given a trajectory with a terminal sparse reward, we first parameterize the score of an action-local region pair with a convolutional neural network F and then aggregate the scores to approximate the final sparse reward. To further enable actionable agents to utilize the scores, a neural dynamics model

---

[1] Project page: https://sites.google.com/view/sapnew/home.

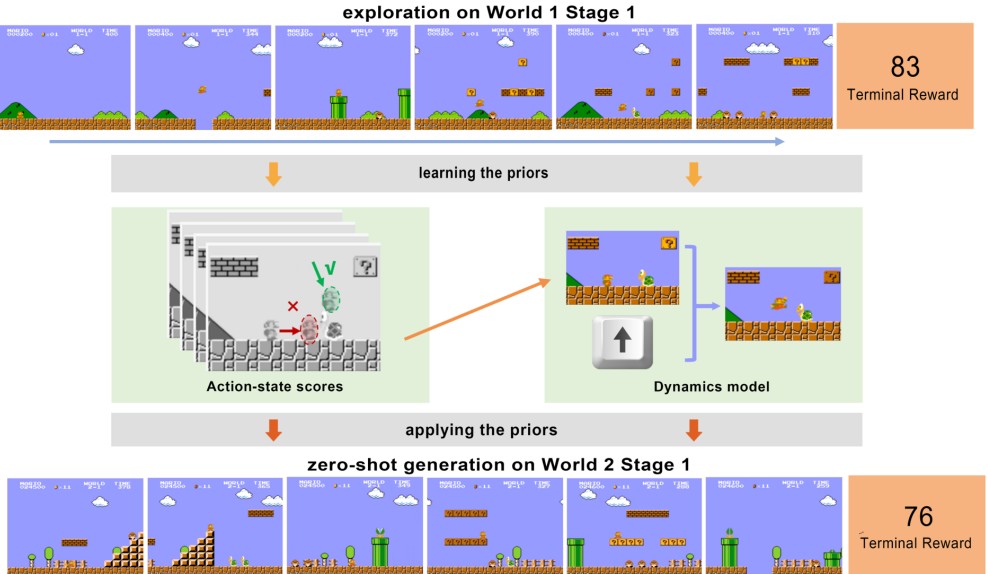

Figure 1: Illustrative figure: An agent is learning priors from exploration data from World 1 Stage 1 in Nintendo Super Mario Bros game. In this paper, the agent focuses on learning two types of priors: learning an action-state preference score for local regions and a dynamics model. The action-state scores on the middle left learns that approaching the "Koopa" from the left is undesirable while from the top is desirable. On the middle right, a dynamics model can be learned to predict a future state based on the current state and action. The agent can apply the priors to a new task World 2 Stage 1 to achieve reasonable policy with zero shot.

can be learned from the interaction data using self-supervision. We show that how an agent can take advantage of the scoring function and the learned dynamics model with planning algorithms (Mayne et al., 2000). We adopt the sparse terminal reward setting because in most of the tasks, step-by-step rewards are hard to obtain while final evaluations for trajectories are relatively easy.

Readers may argue that learning a dense score for every interaction step is reminiscent of Inverse Reinforcement Learning (Ng et al., 2000; Abbeel & Ng, 2004). The distinctions between the proposed method and IRL are threefold: First, instead of learning a reward function of state $s$, we learn a scoring function of a local state $s_l$ and an action $a$, which is sufficiently rich in a physical environment and experimentally can generalize well. Second, with the scoring function in hand, we use a dynamics model learned from passive data to obtain the actual policy in a model-based manner while IRL needs to re-train an agent that can be as data inefficient as model-free RL. However, IRL can have difficulty learning a useful model because the expert demonstrations usually only cover a small portion of the true dynamics. Third, we eliminate the assumption of expensive expert demonstrations with the cost of adding a relatively economical sparse reward in the end. This elimination not only reduces the cost for data collection, but also includes more diverse data to train a robust model.

The proposed scoring function, beyond being a cost function for planning, can also be treated as an indicator of the existence of objects, which affect the evaluation of a trajectory. We empirically evaluate the scores for objects extracted in the context of human priors and hence find the potential of using our method as an unsupervised method for object discovery.

In this paper, we have three major contributions. First, we propose a framework that can learn task-agnostic priors that can generalize to novel tasks. Second, we incorporate a self-supervised learned dynamics model with scoring function to learn a useful policy. Third, we demonstrate the effectiveness of the proposed method on a didactic grid-world example, a well-known video game "Super Mario Bros" and a robotics Blocked-Reach environment and show our method outperforms various baselines. Last but not least, we find that objects emerge from our method in an unsupervised manner, which could be useful to other visual tasks.

## 2 PRELIMINARIES

In this paper, we formulate each environment as an Markov Decision Process (MDP). We represent the MDP by a tuple: $(\mathcal{S}, \mathcal{A}, p, r, \gamma)$, where $\mathcal{S}$ is the state space and $\mathcal{A}$ is the action space. A MDP is fully specified by a state $s \in \mathcal{S}$. A MDP evolves with an action $a \in \mathcal{A}$ by a probability distribution $p(s'|s, a)$. The MDP emits a reward $r(s, a)$ each step. $\gamma \in (0, 1)$ is the discount factor.

Reinforcement learning aims to learn a conditional distribution over the action space given a state $\pi(\cdot|s)$ that maximizes the discounted future rewards:

$$\pi^* = \underset{\pi}{\arg\max} \underset{\substack{a_t \sim \pi(a_t|s_t) \\ s_{t+1} \sim p(s_{t+1}|s_t, a_t)}}{\mathbb{E}} \sum_{t=0}^{\infty} \gamma^t r(s_t, a_t)$$

The state transition probability $p(s'|s, a)$ is treated as unknown in model-free RL problems. Model-based methods explicitly learn a dynamics model $\mathcal{M}(\cdot|s, a)$ that specifies the conditional distribution of the next state given the current state s and action a from environmental interactions.

With an environment model $\mathcal{M}$, one can select an action to rollout with the model recurrently in order to maximize the discounted future reward. One method that approximately finds the optimal action is the Model Predictive Control (MPC) algorithm. It looks ahead for a horizon of $H$ steps and selects an action sequence that maximizes the discounted reward for the future $H$ steps:

$$\underset{a_0, \cdots, a_{H-1}}{\arg\max} \sum_{t=0}^{H-1} \gamma^t r(s_t, a_t)$$

where $s_{t+1} = \mathcal{M}(s_t, a_t)$. To simplify notation, here we assume the environment is deterministic and slightly abuse the notations such that $\mathcal{M}(s_t, a_t)$ returns the next state instead of state distribution. We note that MPC can also use the ground truth environment dynamics $p(\cdot|s, a)$.

## 3 METHOD

### 3.1 PROBLEM FORMULATION

An intelligent agent should be able to learn priors from its past experiences and to generalize to related yet unseen tasks. To facilitate such goals, we formulate the problem as follows:

The agent is first presented with a bank of exploratory trajectories $\{\tau_i\}, i = 1, 2 \cdots N$ collected in a training environment $\mathcal{E}_1 = (\mathcal{S}, \mathcal{A}, p)$. Each $\tau_i$ is a trajectory and $\tau_i = \{(\mathbf{s}_t, \mathbf{a}_t)\}, t = 1, 2 \cdots K_i$. These trajectories are ramdom explorations/interactions with the environment. Instead of specifying the task by per step reward, which is the standard MDP setting, we propose to only evaluate the performance of each trajectory with a terminal evaluation $r(\tau)$ in $\mathcal{E}_1$ when a task $\mathcal{T}$ is given.

At test time, we would like the agent to perform the task $\mathcal{T}$, using zero extra interaction with the new but related environment, represented by $\mathcal{E}_2 = (\mathcal{S}', \mathcal{A}, p')$. We assume the test environment $\mathcal{E}_2$ can have different state distribution and related but different dynamics from the experience environment $\mathcal{E}_1$. However, we assume the actions that an agent could perform stay the same. We evaluate an agent on task $\mathcal{T}$ by giving a single terminal reward $r(\tau_i)$ per trajectory $\tau_i$ to its previous experiences. This reward is only used for evaluation and an agent can never utilize it while carrying off the task. The proposed formulation requires much less information and thus more realistic. In this paper, we focus on locomotion tasks with object interactions, such as Super Mario running with other objects in presence, or a Reacher robot acting with obstacles around.

### 3.2 THE SCORING-AGGREGATING-PLANNING (SAP) FRAMEWORK

We propose the Scoring-Aggregating-Planning framework to solve this problem. As an overview, we propose to learn a per step neural scoring function $\mathrm{F}_\theta$ that scores a sub-region, subset of observation space that surrounds the agent. A local region score is simply a sum over all the sub-regions that are in the local region. The local region scores can be aggregated along the trajectory to approximate

the terminal sparse reward. Meanwhile, a dynamics model $\mathcal{M}_\phi$ is learned to approximate the true transition $p(\cdot|\mathbf{s}, \mathbf{a})$ based on past experience $\mathcal{E}_1$. After both the scoring function and the dynamics model are learned, we perform a Model Predictive Control algorithm to get the final policy on the new environment $\mathcal{E}_2$ for the task $\mathcal{T}$.

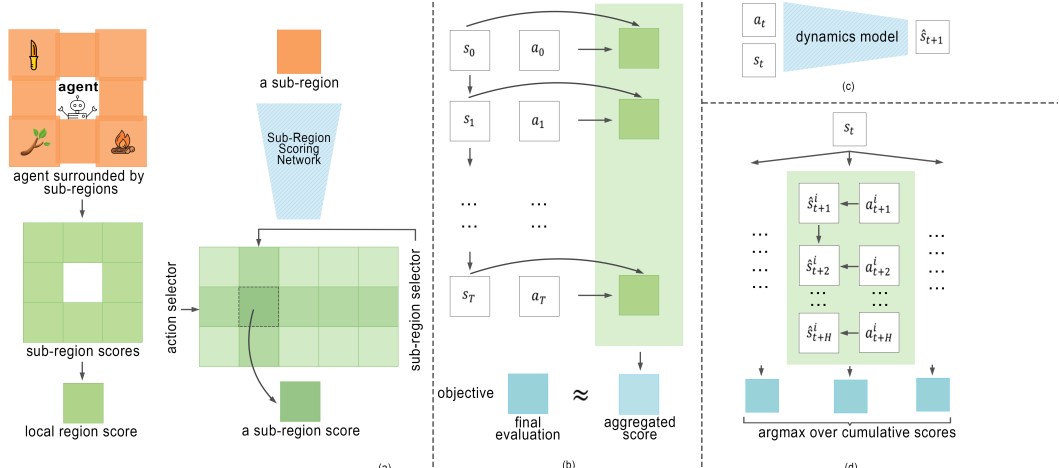

Figure 2: An overview of SAP framework. (a) and (b) in the figure correspond to the Scoring, Aggregating part. (c) and (d) together describes the dynamics learning and planning. See Section 3.2 for details.

**Scoring** The per step sub-region scoring function can be described as $F_\theta(W_l(\mathbf{s}_t), \mathbf{a}_t)$. Here $\theta$ denotes parameters, $W$ is a function that extracts local regions from states and $l$ is a sub-region indicator in a metric space $\mathcal{L}$. We note that the neural network's parameters are shared for every local region. A local region score then is $\sum_{l \in \mathcal{L}} F_\theta(W_l(\mathbf{s}_t), \mathbf{a}_t)$. Intuitively, this function measures how well an action $\mathbf{a}_t$ performs in the current task, based on a local region $l$ extracted from state $\mathbf{s}$.

We presume a local region can be extracted for scoring because in a physical environment, an agent could only interact with the world within its sensing capabilities. From another perspective, it can be seen as a decomposition of the sparse reward into the per-step rewards.

Specifically, on the metric space associated with the problem, we divide the local region around the agent into $n$ sub-regions $l$. For example, in a Cartesian coordinate system, we can divide each coordinate independently. For each $l$, we use a sub-region scoring network to produce a scoring table of size $|\mathcal{A}| \times |\mathcal{L}|$, where $|\mathcal{A}|$ is the action dimension, and $|\mathcal{L}|$ denotes the number of possible relative positions around the agent. One entry is selected from the table as the score for this sub-region $l$ based on the current action taken and the relative position of this sub-region $l$ to the agent.

**Aggregating** To approximate the terminal rewards and learn the scoring network, we aggregate the local region scores $\sum_{l \in \mathcal{L}} F_\theta(W(\mathbf{s}_l), \mathbf{a}_t)$ for each step into a single aggregated score $J$, by an aggregating function $G$:

$$J_\theta(\tau) = G_{(\mathbf{s}_t, \mathbf{a}_t) \in \tau}(\sum_{l \in \mathcal{L}} F_\theta(W_l(\mathbf{s}_t), \mathbf{a}_t))$$

The aggregated score $J$ are then fitted to the sparse terminal reward. In practice, $G$ is chosen based on the form of the final sparse terminal reward, *e.g.* a max or a sum function. In the learning process, the $F_\theta$ function is learned by back-propping errors between the terminal sparse reward and the predicted $J$ through the aggregation function. In this paper, we use $\ell_2$ loss that is: $\min_\theta \frac{1}{2}(J_\theta(\tau) - r(\tau))^2$.

**Planning** To solve the task, we propose to use planning algorithms to find optimal action based on the learned scoring function and a learned dynamics model. As shown in the part (c) of Figure 2, we learn a forward dynamics model $\mathcal{M}_\phi$ based on the exploratory data with a supervised loss function. Specifically, we train a neural network that takes in the action $a_t$, state $s_t$ and output $\hat{s}_{t+1}$, which is an estimate of $s_{t+1}$. We use an $\ell_2$ loss as the objective: $\min_\phi \frac{1}{2}(\mathcal{M}_\phi(s_t, a_t) - s_{t+1})^2$

With the learned dynamics model and the scoring function, we solve an optimization problem using the Model Predictive Control (MPC) algorithm to find the best trajectory for a task $\mathcal{T}$ in environment $\mathcal{E}_2$. The objective of the optimization problem is to minimize $-\tilde{J}_\theta(\tau')$.

Here $\tau'$ is a H-step trajectory sampled based on $\mathcal{M}_\phi$ starting from some state $s_i$. The actual algorithm can be stated as starting from the current state $s_{t=i}$, we randomly select several action sequence up to length $H$. With the dynamics model, we can roll out the estimated states $\hat{s}_{t+1}, \cdots, \hat{s}_{t_H}$. The learned scoring function and the aggregation function can give us an aggregated score for each of the action sequence. We select the action sequence that gives us the best aggregated score and execute the first action in the environment. And we repeat the procedure starting at the new state $s_i$.

## 4 RELATED WORK

**Inverse Reinforcement Learning.** The seminal work Ng et al. (2000) proposed inverse reinforcement learning (IRL). IRL aims to learn a reward function from a set of expert demonstrations. The original IRL is demonstrated on physical state representations, while recent work (Tucker et al., 2018; Rhinehart & Kitani, 2017) has attempted to extend the work into visual states. Although IRL and SAP both learn functions from a set of off-policy data, they fundamentally study different problem — IRL learns a reward function, which can be used for a model-free RL algorithm, from *expert* demonstrations, while our method learns from exploratory data that is not necessarily related to any tasks. There are some works dealing with violation of the assumptions of IRL, such as inaccurate perception of the state (Bogert & Doshi, 2015; Wang et al., 2002; Bogert et al., 2016; Choi & Kim, 2011), or incomplete dynamics model (Syed & Schapire, 2008; Bogert & Doshi, 2015; Levine & Abbeel, 2014; Bagnell et al., 2007; Ng, 2004); however, IRL does not study the case when the dynamics model is purely learned and the demonstrations are suboptimal. Recent work (Xie et al., 2019) proposed to leverage failed demonstrations with model-free IRL to perform grasping tasks; though sharing some intuition, our work is different because of the model-based nature.

**Reward Shaping.** Ng et al. (1999) studied the problem of reward shaping, *i.e.* how to change the form of the reward without affecting the optimal policy. The scoring-aggregating part can also be thought as a novel form of reward shaping where reward functions are automatically learned. Most of the efforts in reward shaping require careful manual design (OpenAI, 2018; Wu & Tian, 2016). A corpus of literature (Marthi, 2007; Grześ & Kudenko, 2010; Marashi et al., 2012) try to learn the reward shaping automatically. Marthi (2007) assumes that the state space can be abstracted, such that one can form an abstracted MDP, which can be solved exactly. Other automatic reward shaping methods, such as Grześ & Kudenko (2010); Marashi et al. (2012), try to build a graph on top of discretized states. However, the methods do not apply to the high-dimensional input such as image, while our SAP framework could. One recent work RUDDER (Arjona-Medina et al., 2018) utilizes an LSTM to decompose rewards into per-step rewards. This method is orthogonal and complementary to our framework.

**RL with Sparse Rewards.** When only sparse rewards are provided, an RL agent suffers a harder exploration problem. In the literature, there are mainly three categories of methods to deal with this problem for high dimensional tasks: (1) Unsupervised exploration strategies, such as curiosity-driven exploration (Pathak et al., 2017; Burda et al., 2018a;b; Schmidhuber, 1991; Stadie et al., 2015; Achiam & Sastry, 2017), or count-based exploration (Tang et al., 2017; Strehl & Littman, 2008; Bellemare et al., 2016; Fu et al., 2017; Ostrovski et al., 2017; Machado et al., 2018; Choshen et al., 2018), solve the sparse reward problem by more efficient explorations. (2) In goal-conditioned tasks, such as pushing an object to some random location, one can use Hindsight Experience Replay (Andrychowicz et al., 2017) to learn from experiences with different goals. (3) More generally, one can define auxiliary tasks to learn a meaningful intermediate representations (Huang et al., 2019a; Silver et al., 2018; Dosovitskiy & Koltun, 2016; Riedmiller et al., 2018; Agarwal et al., 2019). Different from previous methods, we approach this problem by learning a scoring function for each timestep, based on the single terminal reward. This effectively converts the single terminal reward to a set of rich intermediate representations, on top of which we can apply planning algorithms, such as MPC.

**Model-Based RL.** In the planning part of our SAP framework, we train a dynamics model. *i.e.* under the umbrella of model-based algorithms (Sutton, 1991). This idea has been widely studied in the area of robotics (Deisenroth et al., 2013; Deisenroth & Rasmussen, 2011; Morimoto & Atkeson, 2003; Deisenroth et al., 2011). The line of work uses a variety of methods to learn an accurate dynamics

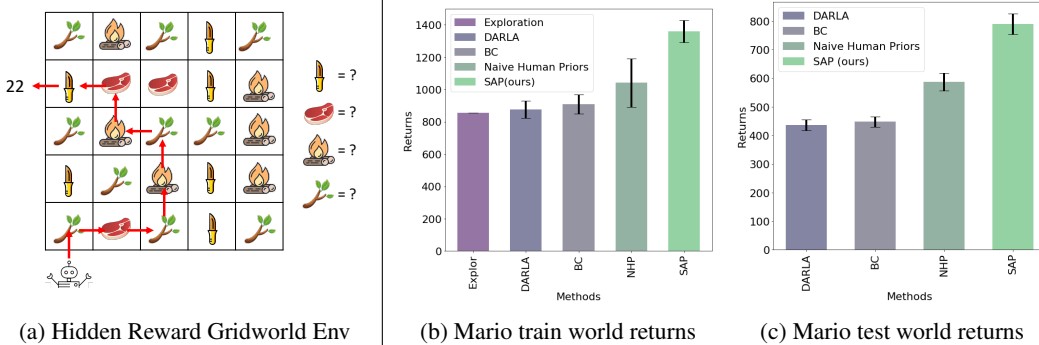

| (a) Hidden Reward Gridworld Env | (b) Mario train world returns | (c) Mario test world returns |

Figure 3: (a) The hidden reward gridworld example. The agent needs to learn the points of each object and collect as many points as possible in a new world configuration. See Section 5.1 for more details. (b) & (c) The total returns of different methods on the Super Mario Bros. All the experiments are run with 50000 game steps on multiple episodes. Error bars are shown as 95% confidence interval except for the given interaction data. See Section 5.2.1 for details. Best view in color.

model ranging from Gaussian Process (Ko & Fox, 2009), time-varying linear models (Levine & Koltun, 2013; Lioutikov et al., 2014; Xie et al., 2016), mixture of gaussian models (Khansari-Zadeh & Billard, 2011) to neural networks (Hunt et al., 1992; Tangkaratt et al., 2014; Kurutach et al., 2018; Chua et al., 2018; Luo et al., 2018; Buckman et al., 2018). This paradigm has been applied to high dimensional space, such as simulated and real robotic applications (Watter et al., 2015; Finn et al., 2016b; Hafner et al., 2018), and Atari games (Kaiser et al., 2019; Weber et al., 2017). Although model-based RL has been extensively studied, none of the previous work has explored combining it with learning the dense task-agnostic scores from sparse signals.

**Zero-Shot Generalization and Priors** Prior knowledge comes from previous experience including interaction with objects, etc. Recently, researchers have shown the importance of priors in playing video games (Dubey et al., 2018). More works have also been done to utilize visual priors such objects in many other domains such as robotics for generalization, etc. (Wang et al., 2019; Jang et al., 2018; Devin et al., 2018; Zhu et al., 2018; Du & Narasimhan, 2019). Keramati et al. (2018); Li et al. (2019); Higgins et al. (2017) explicitly extended RL to handle object level learning. While our method does not explicitly model objects, we have shown that meaningful scores are learned for objects in our SAP framework, which explains why our method generalizes to new tasks without any finetuning. Other works (Sohn et al., 2018; Oh et al., 2017)try to learn compositional skills that can be transferred to new tasks, which is orthogonal and complementary to the proposed method.

## 5 EXPERIMENT

In this section, we would like to study how well our SAP framework performs compare to other methods, and the roles of various components in the framework. We conduct experiments on three environments in different domains: a didactic gridworld task, a video game "Super Mario Bros" (Kauten, 2018) and a robotics blocked reacher task (Huang et al., 2019b). Environment details, architectures, hyper-parameters are described thoroughly in Appendix. A.

### 5.1 DIDACTIC EXAMPLE: HIDDEN REWARD GRIDWORLD

In order to investigate whether the proposed framework can learn meaningful scores and hence induce correct policy, we start with a simple didactic task Hidden Reward Gridworld where the environment matches the assumptions of our method. This environment will reveal to what level can our method recover the per step scores. Figure 3a shows an illustrative example of the Hidden Reward Gridworld. In the grid world, there is an object at each location, where each type of object has some unknown number of points. The agent has already explored some trajectories, and only the sum of points is known by the end of each trajectory. It needs to learn the value of each object and collect as much value as possible in a new environment with different object configurations. In our experiment, we use an $8 \times 8$ grid, with 16 different types of objects, and each of them worth a value of 0, 1 to 15

respectively. To make the task more challenging, instead of giving the identity of each object by object id, we generate a 16-dimensional noisy feature for each type of object. The noisy feature representation of object mimics the output of a perception network from visual inputs.

On this task, our method operates as follows. We use a two layer fully connected neural network to predict per step score from the 16-dimensional feature. Those per step scores are aggregated by a sum operator, and fitted to the training data. We opt to use the environment dynamics model, because it can be perfectly learned by a tabular function. As shown in Table 1, we find that a neural network can fit the object value based on the features with an error 0.04 on a training environment and 0.34 on a new task even with the feature noise presence. To see how well our method performs on this task, we train two behavior cloning agents: an agent imitates the exploratory data (denoted as BC-random) and the other imitates the SAP behavior on the training environment (denoted as BC-SAP). As shown in Table 1, BC-random has far inferior performance, since it clones the exploration behavior, which does not maximize the value collected. BC-SAP performs as well as SAP in the training environment but performs worse than SAP in the new environment. This shows that even allowed to clone the behavior of SAP, behavior cloning still do not generalize as well as our method in a new environment.

Table 1: The average error for the learned object points and the total returns. See Sec. 5.1 for analysis.

|  | World 1 | | World 2 (new task) | |
| --- | --- | --- | --- | --- |
|  | error | returns | error | returns |
| $SAP$ | 0.04 | 378.6 | 0.34 | **389.8** |
| $BC - random$ | - | 83.05 | - | 106.8 |
| $BC - SAP$ | - | - | - | 300.0 |

In this grid world environment, we have shown that SAP is capable of learning accurate scoring functions, even in the presence of object feature noises. We have also demonstrated that SAP would generalize better than alternative algorithms.

## 5.2 SAP ON SUPER MARIO BROS WITH SPARSE REWARDS

To evaluate our proposed algorithm in a more realistic environment, we run SAP and a few alternative methods in the Super Mario Bros environment. In this environment, we neither have a clear definition of the dense score for each step, nor the aggregation procedure. This environment also features high-dimensional visual observations, which is more challenging since we have a larger hypothesis space. The original game has $240 \times 256$ image input and discrete action space with 5 choices. We wrap the environment following Mnih et al. (2015) and what described in Appendix A.2; finally, we obtain a $84 \times 84$ size 4-frame stacked gray-scale stacked observation. The goal for an agent is to survive and go toward the right as far as possible. The environment returns how far the agent goes to the right at the end of a trajectory as the delayed terminal sparse reward.

We apply our SAP framework as follows. We first divide the local region around the agent into eight 12 by 12 pixel sub-regions based on relative position as illustrated in Figure. 7 in the Appendix. Each sub-region is scored by a CNN, which has a final FC layer to output a score matrix. The matrix has the shape dim(action) by dim(relative position), which are 5 and 8 respectively. Then an action selector and a sub-region selector jointly select row corresponding to the agent's action and the column corresponding to the relative position. The sum of all the sub-region scores forms the local region score. Then we minimize the $\ell_2$ loss between the aggregated local region scores along the trajectory and the terminal reward. A dynamics model is also learned by training another CNN. The dynamics model takes in a 30 by 30 size crop around the agent, the agent's location as well a one-hot action vector. Instead of outputting a full generated image, we only predict the future location of the agent recursively. We avoid video predictive models because it suffers the blurry effect when predicting long term future (Lotter et al., 2016; Finn et al., 2016a). We plan with the learned scores and dynamics model with a standard MPC algorithm with random actions that looks ahead 10 steps.

### 5.2.1 COMPARISONS

We compare our method with the following methods:

**Exploration Data** Exploration Data is the data from which we learn the scores, dynamics model and imitate. The data is collected from a suboptimal policy described in Appendix A.2.5. The average reward on this dataset is a baseline for all other methods. This is omitted in new tasks because we only know the performance in the environment where the data is collected.

**Behavioral Cloning** Behavioral Cloning (BC) learns a mapping from a state to an action on the exploration data using supervised learning. We use cross-entropy loss for predicting the actions.

**DARLA (Higgins et al., 2017)** DARLA relies on learning a latent state representation that can be transferred from the training environments to the testing environment. It achieves this goal by obtaining a disentangled representation of the environment's generative factors before learning to act. We use the latent representation as the observations for a behavioral cloning agent.

**Naive Human Priors** Naive Human Priors method (NHP) incorporates model predictive control with predefined naive human priors, which is +1 score if the agent tries to move or jump toward the right and 0 otherwise. NHP replaces the scoring-aggregation step of our method by a manually defined prior. We note that it might be hard to design human priors in other tasks.

In Figure. 3b, the results show that the proposed SAP model outperforms all the baselines with a large margin on the same level they are trained on. We believe that there are two major reasons for BC's unsatisfactory performance: 1. we only have access to the exploration data which is suboptimal for the task. 2. When seeing rare events in the game (the exploration data hardly reach the ending part), it fails to generalize. SAP also outperforms DARLA on both training and generalization tasks. We believe it is because learning disentangled representations on the Mario games is hard, since observations can change dramatically in game settings. Comparing the SAP model with the NHP method, we demonstrate the learned priors can empower an agent with a better understanding of the world thus stronger performance. We show qualitative results in the subsection 5.2.3 to validate that the learned priors contain meaningful scores that leads to better actions. SAP also outperforms all the baselines in an unseen level without any finetuning (Fig. 3c), which proves that it can generalize well.

### 5.2.2  ABLATIVE STUDIES

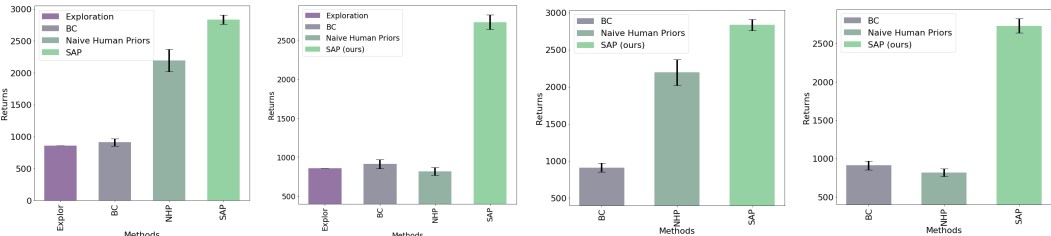

(a) GT dynamics model on W1S1 (train)  (b) GT dynamics model & no done on W1S1 (train)  (c) GT dynamics model on W2S1 (test)  (d) GT dynamics model & no done on W2S1 (test)

Figure 4: Ablative results. (a),(c) are SAP on W1S1 and W2S1 respectively with the groundtruth dynamics model. (b),(d) are similar to (a),(c) but without the true "done" signal. Error bar is 95% confidence interval. We see that with a perfect dynamics model the performance is boosted for both model-based methods. However, even with a disabled "done" signal, SAP still works well while NHP performs significantly worse than before.

Towards understanding the effect of different components of SAP, we investigate the performance of an agent using learned scores on a perfect dynamics model which shows the upper bound for improvement from better models. We further investigate how much the "done" signal from the perfect model help with the performance. Hence, we perform two main ablative experiments:

**Groundtruth Model:** We apply the SAP method with a groundtruth dynamics. Note this setting is only feasible for video games and simulated environments; however, this gives an upper bound for the proposed method by continually improving the dynamics model. In Fig. 4a and Fig. 4c, we find that with a perfect dynamics model, both NHP and SAP has a performance boost on both the original task and a novel task while SAP can still outperform baselines by a large margin. This suggests SAP can perform better if the dynamics model can be further improved.

**No Done Signal:** We hypothesize that the "done" signal from the perfect model contributes to the superior performance because it naturally tells an MPC algorithm not to select a short trajectory if reward at every step is positive (e.g. in the NHP). In Fig. 4b and Fig. 4d, we see that if "done" signal is not provided, SAP still preserves the performance and outperforms all the baselines. However, we find NHP significantly dropped below the level of exploration data return which indicates the NHP heavily relies on the "done" signal.

More ablation studies, such as planning horizon, visual representations in Appendix A.2.7.

### 5.2.3 VISUALIZATION OF LEARNED SCORES AND ACTIONS

In this section, we qualitatively study the induced action by greedily maximizing one-step score. The actions can be suboptimal and different from the policy described above because the greedy actions only consider one-step score but it still capture the behavior of a policy.

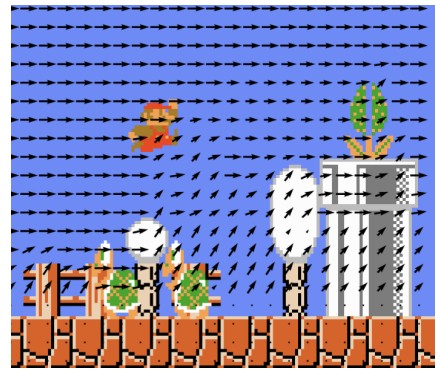

We visualize the computed actions on World 5 Stage 1 (Figure. 5) which is visually different from previous tasks. In this testing case, we see that the actions are reasonable, such as avoiding obstacles and monsters by jumping over them, even in the face of previously unseen configurations and different backgrounds. However, the "Piranha Plants" are not recognized because all the prior scores are learned from W1S1 where it never appears. More visualization of action maps is available in Appendix A.2.8.

Figure 5: Visualized greedy actions from the learned scores in a new environment (W5S1).

Additionally, we visualize the prior scores for different representative local sub-regions in Appendix A.2.8, Figure. 9. In this setting, we synthetically put an agent on a different relative position near an object. We find that our method learns meaningful scores such as assigning low scores for walking towards "koopa"(turtle) and a high score for jumping.

Those qualitative studies further demonstrate that the SAP method can assign meaningful scores for different objects in an unsupervised manner. It also produces good actions even in a new environment.

### 5.3 SAP ON THE 3-D ROBOTICS TASK

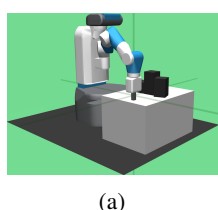 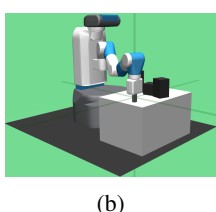 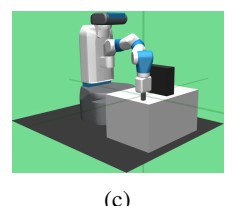 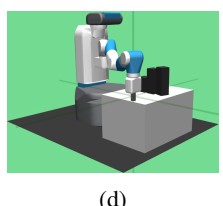

        (a)                    (b)                    (c)                    (d)

Figure 6: Four variants of the 3D robot reacher environments. See Section 5.2.1 for details.

In this section, we further study the SAP method to understand its property with a higher dimensional observation space. We conduct experiments in a 3-D robotics environment, BlockedReacher-v0. In this environment, a robot hand is initialized at the left side of a table and tries to reach the right. Between the robot hand and the goal, there are a few blocks standing as obstacles. The task is moving the robot hand to reach a point on $y = 1.0$ as fast as possible. To test the generalization capability, we create four different configurations of the obstacles, as shown in Figure 6. Figure 6a is the environment where we collect exploration data from and Figure 6b, c, d are the testing environments.

We apply the SAP framework as follows. The original observation is a 25-dimensional continuous state and the action space is a 3-dimensional continuous control. They are discretized into voxels and 8 discrete actions as described in Appendix A.3.1. In this environment, the local region is set to a $15 \times 15 \times 15$ cube of voxels around robot hand end effector. We divide this cube into 27

Table 2: Evaluation of SAP and NHP on the 3D Reacher environment. Numbers are the avg. steps to reach the goal. The lower the better. Numbers in the brackets are the 95% confidence interval.

|  | Config 1 ↓ | Config 2 ↓ | Config 3 ↓ | Config 4 ↓ |
|---|---|---|---|---|
| SAP (w/ learned dynamics) | 97.53[2.48] | 86.53[1.87] | 113.3[3.03] | 109.2[2.98] |
| NHP (w/ learned dynamics) | 124.2[4.51] | 102.0[3.04] | 160.7[7.89] | 155.4[7.24] |
| SAP (w/ perfect dynamics) | 97.60[2.60] | 85.38[1.80] | 112.2[3.07] | 114.1[3.24] |
| NHP (w/ perfect dynamics) | 125.3[4.25] | 102.4[2.96] | 153.4[5.30] | 144.9[4.84] |

$5 \times 5 \times 5$ sub-regions. The scoring function is a fully-connected neural network that takes in a flattened voxel sub-region and outputs the score matrix with a shape of $26 \times 8$. The scores for each step are aggregated by a sum operator along the trajectory. We also train a 3D convolutional neural net as the dynamics model. The dynamics model takes in a $15 \times 15 \times 15$ local region as well as an action, and outputs the next robot hand location. With the learned scores and the dynamics model, we plan using the MPC method with a horizon of 8 steps.

We perform similar baselines as in the previous section that is detailed in Appendix A.3.4. In Table 2, we compare our method with the NHP method on the 3D robot reaching task. We found that our method needs significantly fewer steps than the NHP method, both in the training environment and testing ones. We find that SAP significantly moves faster to the right because it learns a negative score for the obstacles. However, the NHP method, which has +1 positive for each 1 meter moved to the right, would be stuck by the obstacles for a longer duration. We found that the learned dynamics model is relatively accurate in this domain, such that the performance using the learned dynamics is close to that of perfect dynamics. These experiments show that our method can be applied to robotics environment that can be hard for some algorithms due to the 3-D nature. Moreover, we demonstrate again that using partial states (local regions) with SAP generalize better than baselines.

## 6 CONCLUSION

We devise a novel Scoring-Aggregating-Planning (SAP) framework for designing algorithms that can learn generalizable priors from exploration and sparse rewards for novel tasks. We find the proposed method can capture the transferable priors and take advantage of the priors without any finetuning. Experimental results also show that following the SAP framework, designed algorithms outperform a variety of baseline methods on both training and unseen testing tasks in different application domains.

While this paper explores some applications under SAP framework, many compelling questions remain open. There is a welcoming avenue for future work to improve each component of the framework. For example, for complex tasks, there might be priors beyond contact force and game dynamics with a much more complicated action space. Hence, how to extract relational priors from them to solve novel tasks is still yet to be explored. Dubey et al. (2018) thoroughly studied existing human priors in video game playing; however, it is still not clear how to use the correct priors for real-world applications in an SAP framework (*e.g.* fire is good when you want to look but harmful while being too close).

There are many interactive samples in the real world, but most of them are suboptimal. However, an evaluation of them can be given from a task-specific score or human evaluation. Our method excels in this setting. In theory, it can be extended to the case a binary sparse reward is given by carefully choosing an aggregator such as logic operators with sufficient samples. We leave those extensions for future works.

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

## A    EXPERIMENT SPECS

### A.1    HIDDEN REWARD GRIDWORLD

#### A.1.1    ENVIRONMENT.

In Figure 3a, we visualize a sample of the gridworld environment. Each entry correspond to a noised feature vector based on the type of object in it. Each feature is a length 16 vector whose entries are uniformly sampled from $[0, 1]$. Upon each feature, we add a small random noise from a normal distribution with $\mu = 0, \sigma = 0.05$. The outer-most entries correspond to padding objects whose rewards are 0. The action space includes move toward four directions up, down, left, right. If an agent attempts to take an action which leads to outside of our grid, it will be ignored be the environment.

#### A.1.2    ARCHITECTURES FOR SCORE FUNCTION AND DYNAMICS MODEL.

We train a two layer fully connected neural networks with 32 and 16 hidden units respectively and a ReLU activation function to approximate the score for each grid.

In this environment, we do not have a learned dynamics model.

**Hyperparameters.** During training, we use an adam optimizer with learning rate 1e-3, $\beta_1 = 0.9$, $\beta_2 = 0.999$. The learning rate is reduced to 1e-4 after 30000 iterations. The batchsize is 128. We use $horizon = 4$ as our planning horizon.

### A.2    SUPER MARIO BROS

#### A.2.1    ENVIRONMENT.

We wrap the original Super Mario environments with additional wrappers. We wrap the action space into 5 discrete joypad actions, none, walk right, jump right, run right and hyper jump right. We follow (Burda et al., 2018b) to add a sticky action wrapper that repeats the last action with a probability of 20%. Besides this, we follow add the standard wrapper as in past work (Mnih et al., 2015).

#### A.2.2    ARCHITECTURES FOR SCORE FUNCTION AND DYNAMICS MODEL.

For the score function, we train a CNN taking each 12px by 12px sub-region as input with 2 conv layers and 1 hidden fully connected layers. For each conv layer, we use a filter of size 3 by 3 with stride 2 with number of output channels equals to 8 and 16 respectively. "Same padding" is used for each conv layer. The fully connected layers have 128 units. Relu functions are applied as activation except the last layer.

For the dynamics model, we train a neural network with the following inputs: a. 30 by 30 local observation around mario. b. current action along with 3 recent actions encoded in one-hot tensor. c. 3 most recent position shifts. d. one-hot encoding of the current planning step. Input a is encoded with 4 sequential conv layers with kernel size 3 and stride 2. Output channels are 8, 16, 32, 64 respectively. A global max pooling follows the conv layers. Input b, c, d are each encoded with a 64 node fc layer. The encoded results are then concatenated and go through a 128 units hidden fc layer. This layer connects to two output heads, one predicting shift in location and one predicting "done" with sigmoid activation. Relu function is applied as activation for all intermediate layers.

#### A.2.3    HYPERPARAMETERS.

During training, we use an adam optimizer with learning rate 3e-4, $\beta_1 = 0.9$, $\beta_2 = 0.999$. The batchsize is 256 for score function training and 64 for dynamics model. We use $horizon = 10$ as our planning horizon. We use a discount factor $\gamma = 0.95$ and 128 environments in our MPC.

#### A.2.4    MORE ON TRAINING

In the scoring function training, each data point is a tuple of a down sampled trajectory and a calculated score. We down sample the trajectory in the exploration data by taking data from every

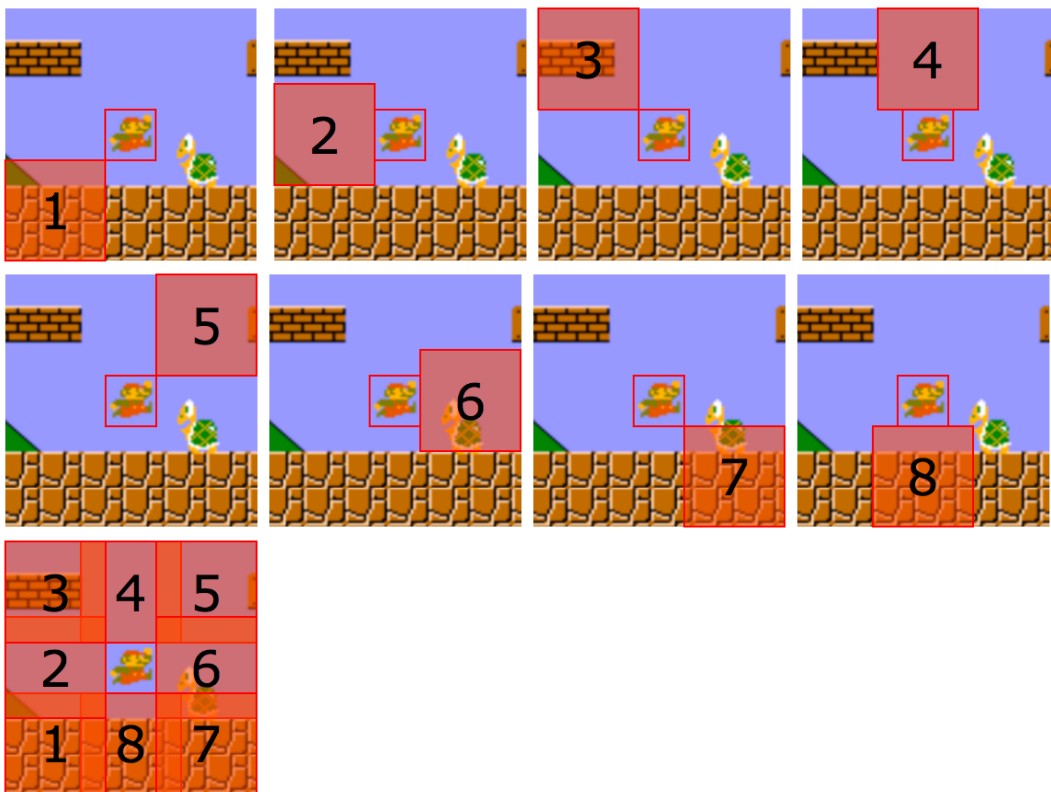

Figure 7: A visualization of the sub-regions in the Super Mario Bros game. In this game, there are in total 8 sub-regions.

Table 3: Ablation Study for number of planning steps in MPC based methods.The averaged return is reported.

| | World 1 Stage 1 | | | World 2 Stage 1(new task) | | |
|---|---|---|---|---|---|---|
| | plan 8 | plan 10 | plan 12 | plan 8 | plan 10 | plan 12 |
| $SAP$ | **1341.5** | **1359.3** | **1333.5** | **724.2** | **790.1** | **682.6** |
| $NHP$ | 1193.8 | 1041.3 | 1112.1 | 546.4 | 587.9 | 463.8 |

Table 4: Ablation Study of visual representation. The averaged return is reported. We omit the baselines that is weaker than NHP.

| | World 1 Stage 1 | World 2 Stage 1(new task) |
|---|---|---|
| $SAP - subregion$ | **1359.3** | **790.1** |
| $SAP - localregion$ | 1258.0 | 737.0 |
| $NHP$ | 1041.3 | 587.9 |

two steps. Half the the trajectories ends with a "done"(death) event and half are not. For those ends with "done", the score is the distance mario traveled by mario at the end. For the other trajectories, the score is the distance mario traveled by the end plus a mean future score. The mean future score of a trajectory is defined to be the average extra distance traveled by longer (in terms of distance) trajectories than our trajectory. We note that all the information are contained in the exploration data.

### A.2.5    MORE DETAILS ON BASELINES

**Behaviroal Cloning (BC).** As super mario is a deterministic environment, we noticed pure behavior cloning trivially get stuck at a tube at the very beginning of level 1-1 and die at an early stage of 2-1. Thus we select action using sampling from output logits instead of taking argmax.

**Exploration Data** We train a policy with only curiosity as rewards (Pathak et al., 2017). However, we early stopped the training after 5e7 steps which is far from the convergence at 1e9 steps. We further added an $\epsilon$-greedy noise when sampling demonstrations with $\epsilon = 0.4$ for 20000 episodes and $\epsilon = 0.2$ for 10000 episodes.

### A.2.6    ADDITIONAL ABLATIONS

**Ablation of Planning Steps** In this section, we conduct additional ablative experiments to evaluate the effect of the planning horizon in a MPC method. In Table. 3, we see that our method fluctuates a little with different planning steps in a relatively small range and outperforms baselines constantly. In the main paper, we choose $horizon = 10$. We find that when plan steps are larger such as 12, the performance does not improve monotonically. This might be due to the difficult to predict long range future with a learned dynamics model.

**Ablation of visual representation** In this section, we conduct experiments to evaluate the effect of the proposed visual representation — local sub-regions. As a comparision, we include a variant that takes in the whole local region as input and output a score conditioned on actions. In Table 4, we see that the local subregions contribute to both the training performance and the zero-shot generalization performance. However, we also find that even without the subregions, SAP still outperforms our second strongest baseline. This is because the scoring and planning steps still empowers the agent the ability to learn and generalize.

**Model Dissection:** To further understand each component of SAP, we ablate the scoring-aggregating component and the planning component. The NHP method uses a manually designed scoring function, with the original planning component. We further conducted the SAP-3 step and the Greedy method which only have 3 planning steps and no planning respectively. In the Table. 5, we observe that without the scoring aggregating component or the planning component, the performance has a significantly drop. This shows that all components of SAP are critical to the performance.

Table 5: Model dissection comparisons. The averaged return is reported. 95% confidence interval is shown in the bracket.

|  | Scoring | Aggregating | Planning | W1S1(train) | W2S1(test) |
|---|---|---|---|---|---|
| SAP | yes | yes | yes | **1359.3[68.5]** | **790.1[35.7]** |
| NHP | manual | no | yes | 1041.3[149.0] | 587.8[31.0] |
| SAP-3 steps | yes | yes | minimal | 944.5[133.5] | 444.3[14.7] |
| Greedy | yes | yes | no | 302.0[27.9] | 40.0[15.7] |

Table 6: Additional baselines privileged BC and RL agent. The averaged return is reported.

|  | World 1 Stage 1 | World 2 Stage 1(new task) |
|---|---|---|
| *SAP* | **1359.3** | **790.1** |
| Privileged BC | 1241.4 | 432.4 |
| RL curiosity | 1183.0 | 347.1 |

### A.2.7 ADDITIONAL BASELINES

In this section, we compare SAP with more baselines. The first baseline is privileged BC: We collected 8000 near-optimal trajectories (average score 1833.0) from the training environment. Then we train an imitation learning agent that mimics the near-optimal data. We note that this baseline is not a fair comparison because SAP only utilizes random exploratory data; we present this baseline to test the generalization ability of an imitative agent that performs well on training environment. The second baseline is a reinforcement learning agent that is trained with curiosity driven reward (Burda et al., 2018b) and the final sparse reward. We limit the training steps to 10M. This is also violating the setting we have as it interacts with the environment. We conduct this experiment to test the generalization ability of a reinforcement learning agent. In Table 6, we see that both baselines have a large drop on the generalization task compared to SAP.

### A.2.8 ADDITIONAL VISUALIZATION

In this section, we present additional visualization for qualitative study. In Figure. 8, we see that on a few randomly sampled frames, even the greedy action can be meaningful for most of the cases. We see the agent intend to jump over obstacles and avoid dangerous monsters.

In Figure. 9, we show the scores of a given state-action pair and find that the scores fulfill the human prior. For example, in Figure. 9a, we synthetically put the mario agent in 8 relative positions to "koopa" conditioned on the action "move right". The score is significantly lower when the agent's position is to the left of "koopa" compared to other position. In Figure. 9b, it is the same setup as in Figure. 9a but conditioned on the action "jump". We find that across Figure. 9a and Figure. 9b the

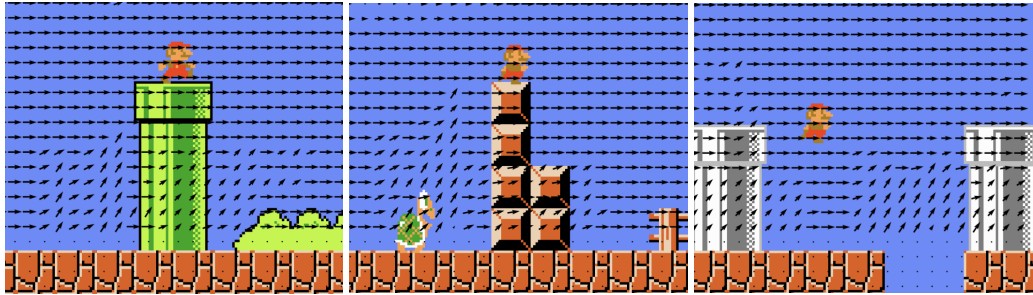

Figure 8: More visualizations on the greedy action map on W1S1, W2S1(new task) and W5S1(new task). Note the actions can be different from the policy from MPC.

left position score of Figure. 9b is smaller than that of Figure. 9a which is consistent with human priors. In Figure. 9c and Figure. 9c, we substitute the object from "koopa" to the ground. We find that on both Figure. 9c and Figure. 9c the score are similar for the top position which means there is not much difference between different actions.

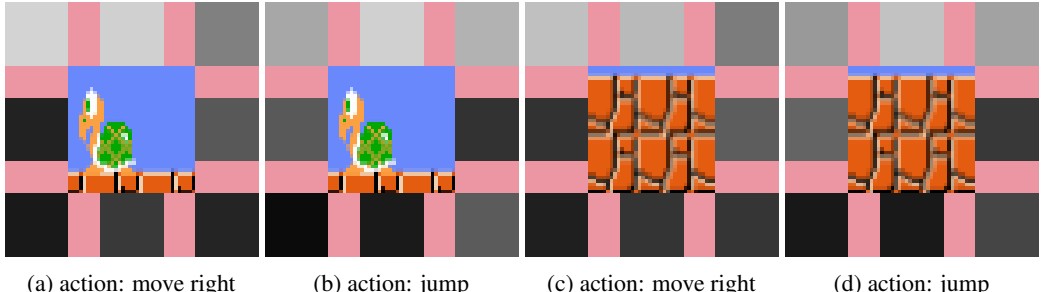

    (a) action: move right      (b) action: jump      (c) action: move right      (d) action: jump

Figure 9: Visualization of the learned score on pre-extracted objects. The grayscale area is the visualized score and the pink area is a separator. Best viewed in color. In (a), we synthetically put the mario agent in 8 relative positions to "koopa" conditioned on the action "move right". The score is significantly lower when the agent's position is to the left of "koopa" compared to other position. In (b), it is the same setup as in (a) but conditioned on the action "jump". We find that across (a) and (b) the left position score of (b) is smaller than that of (a) which is consistent with human priors. In (c) and (d), we substitute the object from "koopa" to the ground. We find that on both (c) and (d) the score are similar for the top position which means there is not much difference between different actions. Note this figure is only for visualizations and we even put the agent in positions that cannot be achieved in the actual game.

### A.3 ROBOTICS BLOCKED REACH

#### A.3.1 ENVIRONMENT.

In the Blocked Reach environment, a 7-DoF robotics arm is manipulated for a specific task. For more details, we refer the readers to Plappert et al. (2018). We discretize the robot world into a $200 \times 200 \times 200$ voxel cube. For the action space, we discretize the actions into two choices for each dimension which are moving 0.5 or -0.5. Hence, in total there are 8 actions. We design four configurations for evaluating different methods as shown in Figure 6. For each configurations, there are three objects are placed in the middle as obstacles. The height of the objects in each configuration are (0.05, 0.1, 0.08), (0.1, 0.05, 0.08), (0.12, 1.12, 0.12), (0.07, 0.11, 0.12).

#### A.3.2 ARCHITECTURES FOR SCORE FUNCTION AND DYNAMICS MODEL.

For the score function, we train a 1 hidden layer fully-connected neural networks with 128 units. We use Relu functions as activation except for the last layer. Note that the input 5 by 5 by 5 voxels are flattened before put into the scoring neural network.

For the dynamics model, we train a 3-D convolution neural network that takes in a local region (voxels), action and last three position changes. The 15 by 15 by 15 local voxels are encoded using three 3d convolution with kernel size 3 and stride 2. Channels of these 3d conv layers are 16, 32, 64, respectively. A 64-unit FC layer is connected to the flattened features after convolution. The action is encoded with one-hot vector connected to a 64-unit FC layer. The last three $\delta$ positions are also encoded with a 64-unit FC layer. The three encoded features are concatenated and go through a 128-unit hidden FC layer and output predicted change in position. All intermediate layers use relu as activation.

#### A.3.3 HYPERPARAMETERS.

During training, we use an adam optimizer with learning rate 3e-4, $\beta_1 = 0.9$, $\beta_2 = 0.999$. The batchsize is 128 for score function training and 64 for dynamics model. We use $horizon = 8$ as our planning horizon.

### A.3.4 BASELINES

Our naive human prior baseline in the blocked robot environment is a 8-step MPC where score for each step is the y component of the action vector at that step.

We omit the Behavioral cloning baselines, which imitates exploration data, as a consequence of two previous results.

