# OpenReview forum: "Scoring-Aggregating-Planning: Learning task-agnostic priors from interactions and sparse rewards for zero-shot generalization"
_ICLR.cc/2020/Conference — Reject_

### Official Review · AnonReviewer1 · 2019-10-28
**Official Blind Review #1**

**Rating:** 6

**Review:**

I am not from this area and don't know much about reinforcement learning.

The paper discusses zero shot generalization (adaptation) into new environments. The authors propose an approach and then show results on Grid-World, Super Mario Bros, and 3D Robotics.

In the training environment E1 = (S, A, p) the algorithm sees a bank of exploratory trajectories \tau_i = {(s_t, a_t)}_{t=1}^{T} but not rewards. The authors  then say that algorithm is tested on the test environment E2. They " propose to only inform the new task per trajectory terminal evaluation r(τ ) in E1" to give the training signal (where r is the reward).

I am a bit confused by this setting. The model never sees any rewards for E1 but it does see rewards for E2? How is this zero shot?

The authors then propose their approach, I wish some of it had been described more rigorously with math (e.g. the loss etc.) so it was easier to understand for people not in the domain and familiar with some of the terminology.

Empirically the authors show results for 3 datasets and this seems thorough.

**Experience Assessment:**

I do not know much about this area.

**Review Assessment: Checking Correctness Of Derivations And Theory:**

I assessed the sensibility of the derivations and theory.

**Review Assessment: Checking Correctness Of Experiments:**

I assessed the sensibility of the experiments.

**Review Assessment: Thoroughness In Paper Reading:**

I read the paper at least twice and used my best judgement in assessing the paper.

---

> ### Author Response · Authors · 2019-11-09
> **Response**
>
> We thank the reviewer for the insightful review and the comments! We address the questions and comments as follows:
>
> —  “I am a bit confused by this setting. The model never sees any rewards for E1 but it does see rewards for E2? How is this zero-shot?”
>
> The agent will not see any reward during the test/generalization stage. In the paper page 3 section 3.1, we illustrate that r(\tau) here is only used for evaluation. Hence, it does not violate the zero-shot setting.
>
> In the original paper, we decompose the data in E1 (training env) into two parts: the trajectory and the reward. We assume the trajectory is collected initially without any reward in mind, and when later we need to perform some task, we specify the task by providing rewards in E1.
>
> Thanks for pointing this out! We will clarify that the reward is hidden from the agent in the testing environments in our revision.
>
> —   “I wish some of it had been described more rigorously with math (e.g. the loss etc.) so it was easier to understand for people not in the domain”
>
> We agree that more rigorous math would help us improve the quality of the paper.  We first clarify the losses defined in our paper here. For the score-aggregation objective, it is $\min_{\theta} \frac{1}{2}(J_\theta(\mathbf{\tau})-r(\tau))^2$ where J is the aggregated score and $r(\tau)$is final evaluation.  The model training objective is $\min_\phi \frac{1}{2}(\mathcal{M}_\phi(\mathrm{s}_t,\mathrm{a}_t)-\mathrm{s}_{t+1})^2$ where $\mathcal{M}$ is a neural dynamics model parameterized by $\phi$, and $s_{t+1}$ is the state in the future timestep. The objective of the Model-Predictive Control is thoroughly described in the preliminary section. We will improve regarding this aspect in our revision.
>
> —  Terminology explanation
>
> We are happy to explain more terminology in the paper! We will slightly expand the preliminary and the related work section to add more explanations.

---

### Official Review · AnonReviewer2 · 2019-10-29
**Official Blind Review #2**

**Rating:** 6

**Review:**

The paper describes a method that aims to learn task-agnostic priors for zero-shot generalization. The main idea is to employ the following modeling approach on top of the model-based RL framework: a local convolution network is used to compute a score for each local state action pair, and then another network is used to aggregate all the scores. While the problem being studied is important and the experimental results seem positive, there are a few concerns.

First, the baselines presented in the experiments are relatively weak. In Related Work, the authors discuss the differences between the proposed method and the related methods, but few of the related methods are used as baselines for comparison with the proposed method. Moreover, the experiments are quite insufficient in terms of ablating different components of the proposed methods.

Second, essentially the proposed method is trying to solve the zero-shot generalization by parameter initialization; a model is pretrained on related tasks and used as initializations for target tasks. The authors claim that it is different from prior work mainly because of the neural architecture that deals with sparse rewards via score aggregation. While the proposed architecture might be more suitable for solving tasks with sparse rewards, it is not intuitive whether it has something to do with learning zero-shot generalization. And apparently, the method will also rely on the similarity between the pretrained task and the target task, and such a scope constraint is not discussed in the paper. In other words, I'm not quite sure a better architecture is fundamental progress towards zero-shot RL.

**Experience Assessment:**

I do not know much about this area.

**Review Assessment: Checking Correctness Of Derivations And Theory:**

N/A

**Review Assessment: Checking Correctness Of Experiments:**

I assessed the sensibility of the experiments.

**Review Assessment: Thoroughness In Paper Reading:**

I read the paper at least twice and used my best judgement in assessing the paper.

---

> ### Author Response · Authors · 2019-11-09
> **Response - part (1/2)**
>
> We thank the reviewer for the insightful review and comments on our work! We address the questions and comments:
>
> “First, the baselines presented in the experiments are relatively weak. In Related Work, the authors discuss the differences between the proposed method and the related methods, but few of the related methods are used as baselines for comparison with the proposed method. ”
>
> There are mainly three related domains: zero-shot policy generalization, inverse reinforcement learning, and better architecture design.
>
>
> Some works attacked zero-shot learning in computer vision where the main challenge is to transfer knowledge between different *visual* domains. However, these methods can hardly be any stronger than the proposed baselines in the paper because they did not have any ability to adapt an actionable policy and further deal with different configurations between environments. Another line of work researches mainly on how to leverage the compositionality of trajectories in grid-world like environments so that the skills can be transferred in zero-shot in an unseen world. However, these methods all have strong restrictions about the simulated world being near-discrete (although it can be in a 3D navigation environment) and containing explicit objects with their corresponding concepts.
>
> We did not find any existing Inverse RL baselines that can be applied to the zero-shot with a suboptimal demonstration setting.  Most of IRL methods assume expert demonstration. It then optimizes the objective for learning a family of reward function, with which the demonstrations are superior to any other policies. Moreover, the model-based nature of our method does not require any further interaction with environments while IRL requires *extensive interactions*. We also discussed this in our related work.
>
> To demonstrate the effectiveness of our method, we take the closest work, DARLA (DisentAngled Representation Learning Agent)[1] and adapt to our setting. DARLA is a method designed to zero-shot generalize to novel environments with complex visual inputs. It achieves the generalization by learning to see before learning to act with a disentangled representation from a beta-Variational Autoencoder ($\beta-VAE$) followed by a Denoising Autoencoder (DAE). We compare DARLA with our method and will incorporate this in our next revision:
>
> We note that for Mario, *higher is better*. For robotic control, *lower is better*.
>                       |Mario-Train| Mario-Test |
> | Explore     | 856.7           | N/A            |
> +---------------+----------------+----------------+
> | BC              | 910.2          | 447.8           |
> +---------------+----------------+----------------+
> | NHP          | 1041.3         | 587.8          |
> +---------------+----------------+----------------+
> | *DARLA*  | 876.7          | 436.5           |
> +---------------+----------------+----------------+
> | SAP (ours)| 1359.3        | 790.1           |
>
>
> For comparison between architectures that take in local-subregion-based observations and global observations, we believe that the BC baseline that looks at the global image performs worse than SAP on both the training task as well as the generalization task. This proves that the proposed architecture in SAP with local observations is superior to simply use global observations.
>
> We will incorporate the comparison and more discussion in our next revision.
>
>
> --“Moreover, the experiments are quite insufficient in terms of ablating different components of the proposed methods.”
>
> We conducted experiments for ablative study including the effect of 1) dynamic models, 2) ``"done" signal at the end of an episode, and 3) the number of planning horizons.
>
> In addition to that, we also compare it with another baseline that uses rudder[2] as the aggregator.  This baseline shows that even the proposed summation aggregator is very simple, it is surprisingly more effective than RUDDER aggregator. This might be due to the long horizon of the proposed tasks because RUDDER is designed to fit atari games. Quantitative results are as follows:
>
>                                      | Mario-Train | Mario-Test |
> | Explore                    | 856.7             | N/A             |
> +--------------------------+------------------+----------------+
> | BC                            | 910.2             | 447.8           |
> +--------------------------+------------------+----------------+
> | NHP                         | 1041.3           | 587.8          |
> +--------------------------+------------------+----------------+
> | *Rudder Variant* |  314               | 306              |
> +--------------------------+------------------+----------------+
> | SAP (ours)              | 1359.3           | 790.1           |
>
>  We will add the results to the next revision. Beyond this, we will also modify our existing discussion about the ablative study including the effect of dynamic models, done signal, and planning steps.

---

> ### Author Response · Authors · 2019-11-09
> **Response - part (2/2)**
>
>
> —-” it is not intuitive whether it has something to do with learning zero-shot generalization”
>
> The proposed architecture learns to decompose the final evaluation into subregion- action pair scores. This score is zero-shot transferable when the visual similarity between subregions holds. This is intuitive that visually similar subregions will share similar scores across different configurations. Previous works focus on learning transferable skills between domains, while our method focuses on transferable subregions in the observation space. Empirically, we find such improvement can lead to zero-shot behavioral transfer with a learned dynamics model.
>
>
> —- “The method will also rely on the similarity between the pretrained task and the target task, and such a scope constraint is not discussed in the paper. not quite sure a better architecture is fundamental progress towards zero-shot RL.”
>
> The reviewer questions whether a better architecture is a fundamental progress towards zero-shot RL. We think that the answer is yes, and at the same time, we admit and agree that the success of zero-shot reinforcement learning might also depend on other innovations, such as data representations, compositional skills, etc. This has been proved in the image classification task. People found that convolution neural network is significantly better than the previous hand-designed SIFT + Fisher vector methods. The input and output haven't changed, but only the architecture of the learner has changed. These architectural improvements have also brought us better generalization, including zero-shot generalization in the image classification domain. We believe that architectural improvements will also be fundamental in policy generalizations as well. However, incorporating SAP with other innovations such as compositional skills can be a promising future direction.
>
> You are also correct about the similarity assumption; however, this similarity is texture/color/instance level visual similarity rather than task-level similarity. Even with this assumption, the tasks still leave the generalization between different configurations burden to the learning algorithm. In Dubey, Rachit, et al. "Investigating human priors for playing video games." arXiv preprint arXiv:1802.10217 (2018). , they thoroughly described different types of priors in video games including visual similarity.
>
> [1] Higgins, Irina, et al. "Darla: Improving zero-shot transfer in reinforcement learning." Proceedings of the 34th International Conference on Machine Learning-Volume 70. JMLR. org, 2017.
> [2]Arjona-Medina, Jose A., et al. "Rudder: Return decomposition for delayed rewards." arXiv preprint arXiv:1806.07857 (2018).
> [3] Burda, Yuri, et al. "Exploration by random network distillation." arXiv preprint arXiv:1810.12894 (2018).

---

### Official Review · AnonReviewer3 · 2019-11-03
**Official Blind Review #3**

**Rating:** 3

**Review:**

The paper proposes a framework (Scoring-Aggregating-Planning (SAP)) for learning task-agnostic priors that allow generalization to new tasks without finetuning. The motivation for this is very clear - humans can perform much better than machines in zero-shot conditions because humans have learned priors about objects, semantics, physics, etc. This is achieved by learning a scoring function based on the final reward and a self-supervised learned dynamics model.

Overall, the paper is very clear and easy to follow.
The presented task is realistic and important, and the paper seems to address it in a reasonable approach.
However, the evaluation seems lacking to me - the evaluation convinced me that SAP works, but I am not convinced that it works better than existing approaches (see below), and especially did not convince me that it is better in the zero-shot test environment.
The (anonymized) website contains nice videos that support the submission.

Questions for the authors:

1. Page 3, 3rd paragraph of Section 3: the paper says that "The proposed formulation requires much less information and thus more realistic and feasible" - I agree that this is more realistic, but is it really more feasible? The requirement of much less information makes the proposed formulation much more sparse.

2. A basic assumption in the SAP framework is that a local region score is a sum of all the sub-regions. As phrased in the paper: "in the physical world, there is usually some level of rotational or transnational invariance". I'm not sure that this assumption makes sense neither in the Mario case or in other tasks, e.g., robotics. Doesn't it matter if you have a "turtle" right in front of you (which means that the turtle is going to hit you), or below you (which means that you are going hit the turtle)?

3. A question about the planning phase - page 5 says: "We select the action sequence that gives us the best-aggregated score and execute the first action". Do you select the entire sequence of actions in the new environment in advance? Can the agent observe the new state after every action, and decide on the next action based on the actual step that the action has reached, rather than on the state that was approximated in advance?
In other words - what happens if the first action in the new test environment yields an unexpected state, that was not predicted well by the dynamics model; does the agent continue on the initial planned trajectory (that ignores the "surprise"), or does it compute its next action based on the unexpected state?

4. Experiments: in Gridworld and Mario - are there any stronger baselines in the literature, or reductions of known baselines to the zero-shot scenario? Are the chosen "Human Priors", BC-random and BC-SAP just strawmen?
Since the main goal of this paper is the zero-shot task, what would convince me is a state-of-the-art model that does possibly *better than SAP on the training level*, but *worse than SAP in generalizing to the new level*. Additionally, are there other baselines that specifically address the zero-shot task in the literature?

Minor (did not impact score):
Page 2, 1st paragraph: "... we show that how an intelligent agent"...
Page 3, 3rd paragraph: "... in model-free RL problem" - missing an "a" or "problem*s*"?
Page 3, 3rd paragraph: ". Model based method ..." - missing an "a" as well?
Page 4, 1st paragraph:: "... utilizing the to get the ..."
Page 4, last row: missing a dot after the loss equation, before the word "In".
Page 7, Table 1: "BC-random" is called "BC-data" in the text. Aren't they the same thing?


**Experience Assessment:**

I do not know much about this area.

**Review Assessment: Checking Correctness Of Derivations And Theory:**

I carefully checked the derivations and theory.

**Review Assessment: Checking Correctness Of Experiments:**

I carefully checked the experiments.

**Review Assessment: Thoroughness In Paper Reading:**

I read the paper thoroughly.

---

> ### Author Response · Authors · 2019-11-09
> **Response - part (1/2)**
>
> We thank the reviewer for the insightful review and comments on our work! We address the questions and comments:
>
> — “Page 3, 3rd paragraph of Section 3: the paper says that "The proposed formulation requires much less information and thus more realistic and feasible" - I agree that this is more realistic, but is it really more feasible? The requirement of much less information makes the proposed formulation much more sparse.”
>
> When the formulation is realistic, it can be easily applied to more general real-world problems; more complex formulations usually requires to engineer the original problem. This is how we interpret “feasible”. We will rephrase and clarify this in the revision.
>
> — “A basic assumption in the SAP framework is that a local region score is a sum of all the sub-regions. As phrased in the paper: "in the physical world, there is usually some level of rotational or transnational invariance". I'm not sure that this assumption makes sense neither in the Mario case or in other tasks, e.g., robotics. Doesn't it matter if you have a "turtle" right in front of you, or below you?”
>
> A turtle below Mario and a turtle in front of Mario are different indeed! We clarify that the “rotational or translational invariance” is referring the pixels inside one subregion meaning that when there is some small translational shift/rotational shift inside of one subregion, the scoring network can still give a reasonable output. We do consider the difference between different locations among subregions in the paper. And from Appendix. A.2.7, Figure 9, we can see that the score for a turtle in front is much lower than a turtle below Mario. This reassures your point! We will update the text and clarify this in the next revision.
>
> — “A question about the planning phase - page 5 says: "We select the action sequence that gives us the best-aggregated score and execute the first action". ... In other words - what happens if the first action in the new test environment yields an unexpected state, that was not predicted well by the dynamics model; does the agent continue on the initial planned trajectory (that ignores the "surprise"), or does it compute its next action based on the unexpected state?”
>
> Yes, the planning method used in this SAP is adaptive if it meets an unexpected state. The extract model-predictive control algorithm can be described as:  We select the action sequence that gives us the best-aggregated score based on the learned dynamics and executes the first action in the real environment. Then it reaches a new state which can be similar or very different from the predicted state. Then we redo everything again until the end of an episode.  Thus, we conclude that the planning algorithm will plan based on an unexpected state rather than stick to a predicted one.
>
> — “Are there any stronger baselines in the literature, or reductions of known baselines to the zero-shot scenario?”
>
> For zero-shot learning in computer vision, the main challenge is to transfer knowledge between different visual domains. However, these methods can hardly be any stronger than the proposed baselines in the paper because they did not have any ability to adapt an actionable policy and further deal with different configurations between environments. Another line of work researches mainly on how to leverage the compositionality of trajectories in grid-world like environments so that the skills can be transferred in zero-shot in an unseen world. However, these methods all have strong restrictions about the world containing explicit objects and their corresponding concepts.
>
> We find the closest and state-of-art work[1], which can be reformulated to the setting in our paper and perform zero-shot transfer between domains. The author proposed a new multi-stage RL agent, DARLA (Disentangled Representation Learning Agent), which learns to see before learning to act by learning a disentangle representation from a beta-Variational Autoencoder ($\beta$-VAE) followed by a Denoising Autoencoder (DAE). By decomposing the vision and action, DARLA has achieved state-of-art results in a variety of RL environments and RL algorithms. We compare DARLA with our method (and will be added to our next revision):
>
> We note that for the task, *higher is better*.
>
>                    | Mario-Train | Mario-Test |
> | Explor     | 856.7             | N/A            |
> +-------------+------------------+----------------+
> | BC           | 910.2             | 447.8          |
> +-------------+------------------+----------------+
> | NHP        | 1041.3          | 587.8           |
> +-------------+------------------+---------------+
> | *DARLA*| 876.7            | 436.5          |
> +-------------+------------------+---------------+
> | SAP(ours)| 1359.3         | 790.1          |
>
> We also will incorporate this into our paper in the revision.

---

> ### Author Response · Authors · 2019-11-09
> **Response - part (2/2)**
>
> —-  “better than SAP on the training level*, but *worse than SAP in generalizing to the new level*”
>
> SAP outperforms all the baselines including DARLA on both training and testing environments. This is because SAP is robust to *suboptimal demonstrations* while most existing works cannot handle this. However, to stack the card against SAP and to be more convincing, we follow the request and conduct experiments where the other method performs well on the training level:
>
> We collected 8000 near-optimal trajectories from the training environment. Then we train an imitation learning agent that mimics the near-optimal data. We call this method “privileged BC” because it imitates the data that achieves average return 1833 on the training task.  The other method we compare is a curiosity-driven reinforcement learning method [3] that is trained with PPO on the training task. This method also has a certain level of generalization ability as claimed in the original paper.
>
> The table below shows that in the training environment, both methods achieve similar performance compared to SAP; however, they significantly drop on the testing environment. We attribute the failure of them to the overfitting effect of supervised learning/reinforcement learning.  We also note again, this experiment favors the baseline (since the imitative method uses near-optimal trajectories). It is an ablative study to check the “zero-shot” ability between different methods.  Quantitative results are as follows:
>
> |                      | Mario-Train | Mario-Test |
> +-----------------+-----------------+-----------------+
> | Exploration | 856.7           |       N/A        |
> +-----------------+-----------------+-----------------+
> | BC                | 910.2            | 447.8            |
> +-----------------+-----------------+-----------------+
> | NHP             | 1041.3          | 587.8           |
> +-----------------+-----------------+-----------------+
> | *Priv BC*    |  1241.4        | 432.4            |
> +-----------------+-----------------+-----------------+
> | *curiosity*  | 1183            |  347              |
> +-----------------+-----------------+-----------------+
> | SAP (ours)   | 1359.3         | 790.1            |
> +-----------------+-----------------+-----------------+
>
> We will incorporate this into the next revision of our paper.
>
> —- “Are the chosen "Human Priors", BC-random and BC-SAP just strawmen?”
>
> The proposed baselines are strong baselines, especially the human prior baseline. The “Naive Human Prior” baseline is a reward that is designed manually based on human priors about the environment. This is the routine way people deal with a sparse reward environment. SAP outperforms this mainly because the learned scores are more informative than the manually designed rewards; hence robust even under a learned dynamics model. As for BC baseline, this is generally trying to show that compared with the SAP method, a parameterized policy can hardly compete on unseen tasks.  We will add a discussion in the next revision.
>
>
> Minor:
> Minor (did not impact score):
> Page 2, 1st paragraph: "... we show that how an intelligent agent"...
> Page 3, 3rd paragraph: "... in model-free RL problem" - missing an "a" or "problem*s*"?
> Page 3, 3rd paragraph: ". Model based method ..." - missing an "a" as well?
> Page 4, 1st paragraph:: "... utilizing the to get the ..."
> Page 4, last row: missing a dot after the loss equation, before the word "In".
> Page 7, Table 1: "BC-random" is called "BC-data" in the text. Aren't they the same thing?
>
> Thank you for pointing out the typos.  We will revise the paper to address them.
>
> [1] Higgins, Irina, et al. "Darla: Improving zero-shot transfer in reinforcement learning." Proceedings of the 34th International Conference on Machine Learning-Volume 70. JMLR. org, 2017.
> [2]Arjona-Medina, Jose A., et al. "Rudder: Return decomposition for delayed rewards." arXiv preprint arXiv:1806.07857 (2018).
> [3] Burda, Yuri, et al. "Exploration by random network distillation." arXiv preprint arXiv:1810.12894 (2018).

---

### Decision · Program_Chairs · 2019-12-19

**Decision:**

Reject

**Comment:**

The paper proposes an algorithm for zero-shot generalization in RL via learning a scoring a function from.

The reviewers had mixed feelings, and many were not from the area. A shared theme was doubts about the significance of the experimental setting, and also the generality of the approach.

As this is my field, I read the paper, and recommend rejection at this time. The proposed method is quite laborious and requires quite a bit of assumptions on the environments to work, as well as fine tuning parameters for each considered task (number of regions, etc). I also agree that the evaluation is not convincing -- stronger baselines need to be considered and the experiments to better address the zero-shot transfer aspect that the paper is motivated by. I encourage the authors to take the review feedback into account and submit a future version to another venue.